# Copy Number Variations in Children with Tourette Syndrome: Systematic Investigation in a Clinical Setting

**DOI:** 10.3390/genes14020500

**Published:** 2023-02-15

**Authors:** Federica Saia, Adriana Prato, Lucia Saccuzzo, Francesca Madia, Rita Barone, Marco Fichera, Renata Rizzo

**Affiliations:** 1Child and Adolescent Neurology and Psychiatric Section, Department of Clinical and Experimental Medicine, Catania University, 95124 Catania, Italy; 2Department of Cognitive Sciences, Psychology, Education and Cultural Studies, University of Messina, 98121 Messina, Italy; 3Department of Biomedical and Biotechnological Sciences, Medical Genetics, University of Catania, 95124 Catania, Italy; 4Laboratory of Neurogenetics and Neuroscience, IRCCS Istituto Giannina Gaslini, 16147 Genoa, Italy; 5Research Unit of Rare Diseases and Neurodevelopmental Disorders, Oasi Research Institute-IRCCS, 94018 Troina, Italy

**Keywords:** Tourette Syndrome, CNV, dysmorphism, genetic analysis, tic severity

## Abstract

Tourette syndrome (TS) is a neurodevelopmental disturbance with heterogeneous and not completely known etiology. Clinical and molecular appraisal of affected patients is mandatory for outcome amelioration. The current study aimed to understand the molecular bases underpinning TS in a vast cohort of pediatric patients with TS. Molecular analyses included array-CGH analyses. The primary goal was to define the neurobehavioral phenotype of patients with or without pathogenic copy number variations (CNVs). Moreover, we compared the CNVs with CNVs described in the literature in neuropsychiatric disorders, including TS, to describe an effective clinical and molecular characterization of patients for prognostic purposes and for correctly taking charge. Moreover, this study showed that rare deletions and duplications focusing attention on significant genes for neurodevelopment had a statistically higher occurrence in children with tics and additional comorbidities. In our cohort, we determined an incidence of potentially causative CNVs of about 12%, in line with other literature studies. Clearly, further studies are needed to delineate the genetic background of patients with tic disorders in a superior way to elucidate the complex genetic architecture of these disorders, to describe the outcome, and to identify new possible therapeutic targets.

## 1. Introduction

Tourette syndrome (TS) is a neurodevelopmental disorder, characterized by the existence of numerous motor tics and at least one vocal tic persisting more than one year [1]. Recent estimates suggest that prevalence in the worldwide population ranges between 0.3% and 1% with a male to female ratio of 3–4:1 [2,3,4].

TS has a complex and multifactorial etiology. The susceptibility of TS is based on genetic and environmental factors that play a role together [5,6]. Several studies suggest that TS is one of the neurodevelopmental disorders with the highest rate of non-Mendelian heredity [4]. A polygenic inherited nature for TS is supported by studies on twins [7] and families [8]. Several potential susceptibility genes have been suggested. The involvement of SLIT and NTRK-like family member 1 (SLITRK1) genes has been the subject of numerous debates [9]. The disclosure of mutations on the gene encoding 1-histidine decarboxylase (HDC), an enzyme involved in histamine biosynthesis and, consequently, in the synthesis of dopamine and serotonin, in a multiplex family suggested the hypothesis that histaminergic neurotransmission was implicated in the genesis of TS [10]. A large, genome-wide study involving 1285 patients and 4964 healthy controls showed no significant genetic variants except for the gene *COL27A1*, coding for α1-collagen [11]. Subsequent studies discovered the involvement of other genes, such as *NTN4* (involved in axonal driving processes and in the development of the striated body), *NRXN1* (coding for neurexin 1), and *CNTN6* (coding for contactin 6) [12,13]. In another study, Rizzo et al. (2015) also described the expression in serum of 754 miRNAs in TS patients and found that miR-429 is significantly under-expressed in patients affected by TS with respect to healthy controls [14]. Even with this recognized genetic component, the identification of TS susceptibility genes has been challenging. Rare copy number variants (CNVs) have been demonstrated to represent weighty risk factors for different neuropsychiatric diseases [15,16], including TS [17,18,19]. In accord with this observation, in this study we aimed to evaluate whether the occurrence of potentially causative CNVs (PC-CNVs) was related with frequent clinical features in a cohort of phenotypically well-characterized TS patients. We focused on the incidence of dysmorphic features, epilepsy, brain magnetic resonance imaging (MRI) anomalies, and gravity of symptoms in children with TS and PC-CNVs with respect to those with non-causative CNVs (NC-CNVs) or without CNVs (W-CNVs).

## 2. Materials and Methods

### 2.1. Participants

This study was directed and realized at the Child and Adolescent Neurology and Psychiatry of the Medical and Experimental Department, Catania University. A total of 93 patients with a clinical diagnosis of TS according to DSM-5 criteria, were enrolled for the study between April 2021 and April 2022. We excluded patients with other primary psychiatric disorders different from TS and/or the presence of a specific etiology, such as genetic or metabolic diseases. Each participant was clinically assessed for tics and associated comorbidities. Investigations were performed as part of the routine clinical care of the patients in accordance with the ethical standards laid down in the 1964 Declaration of Helsinki and its later amendments. Prior to enrolment, written informed consent was signed from all participants’ parents or legal guardians. The study was allowed by the local Ethics Committee at our institution.

### 2.2. Procedures

Medical history was acquired from the participants’ parents, giving priority to developmental delays, epilepsy, neuropsychiatric disorders, and positive family history for neurodevelopmental disorders and metabolic diseases. Each participant was subjected to a physical and neurological examination. Fasting blood samples and urine samples were collected for routine blood analyses and to exclude metabolic diseases. The incidence of epileptic seizures and isolated electroencephalogram (EEG) anomalies were examined.

Brain MRI results, measured using a 1.5 T MRI scanner, were used to unmask morphologically visible signs of modified brain development.

### 2.3. Clinical Assessment

All patients were screened with the Schedule for Affective Disorders and Schizophrenia for School-Age Children—Present and Lifetime (Kiddie-SADS-PL) to evaluate other possible comorbidities [20]. Then, all subjects underwent neuropsychiatric evaluation for TS and related comorbidities. Symptoms of TS were evaluated by applying the Yale Global Tic Severity Rating Scale (YGTSS) [21], a clinician-rated scale used to establish the motor and phonic tic severity in light of the number, frequency, duration, intensity, and complexity of tics. To evaluate obsessive compulsive disorder (OCD), typically associated with TS or CTD, the CY-BOCS, a semi-structured clinician-administered interview assessing the severity of obsessions and compulsions appearing over the past week through five areas (time, interference, distressing nature, effort to resist, and control over obsessions and compulsions) was also administered [22]. Regarding ADHD symptoms, patients were also assessed according to the ADHD rating scale [23] and the Conners’ Parent Rating Scale (CPRS). The CPRS is a helpful instrument for acquiring parental reports of childhood behavior problems that contains summary scales supporting ADHD diagnosis and estimating ADHD severity [24]. Diagnosis of ASD was evaluated using gold-standard standardized diagnostic tests, including the Autism Diagnostic Interview-Revised (ADI-R) and Autism Diagnostic Observation Schedule (ADOS). ADI-R is an investigator-based parent or caregiver interview that gives a clear description of history, and at same time of the current functioning, detecting development areas associated with autism [25]. ADOS is a semi-structured, standardized assessment of social affect, which comprehends language, communication and social reciprocal interaction, and restricted and repetitive behaviors for individuals suspected of being affected by ASD [26]. Moreover, all participants completed the MASC, a self-report scale that investigates the presence of anxiety in children aged 8–18 years [27] and the Child Depression Inventory, a 27-item self-report instrument that explores depressive symptoms in 7- to 17-year-olds [28]. Furthermore, all participants were subjected to a full neuropsychiatric assessment. The Wechsler Intelligence Scale for Children (WISC-IV) was administered, according to patient’s age, to establish the intelligence quotient (IQ) of patients [29].

### 2.4. Genetic Analysis

Array-CGH analyses were achieved by applying the Human Genome Array-CGH 8x60K Microarray (Agilent Technologies, Palo Alto, CA, USA), with a typical probe spacing of around 55 Kb. The arrays were performed using Agilent Reference DNAs, analyzed with the Agilent Microarray Scanner Feature Extraction Software v11.5, and Agilent Genomic Workbench 7.0.4.0 software using the ADM-2 algorithm. Genomic positions of the rearrangements refer to the public UCSC database GRCh37/hg19. Inheritance of non-polymorphic CNVs was evaluated with multiplex ligation-dependent probe amplification or real-time PCR assays.

### 2.5. Array CGH Data Analysis and CNVs Classification

“Potentially causative” deletions/duplications, “non-causative”, or potentially benign familiar variants were demonstrated based on the scientific literature and on the accurate consultation of public and private databases, such as the Database of Genomic Variants (http://dgv.tcag.ca/dgv/app/home, accessed on 5 November 2022), the Database of Human CNVs hosted by IRCCS Oasi Maria SS of Troina, (http://gvarianti.homelinux.net/gvariantib37/index.php, accessed on 5 November 2022), the Decipher Database (https://decipher.sanger.ac.uk/, accessed on 5 November 2022), the Database of Human Genomic Structural Variation (https://www.ncbi.nlm.nih.gov/dbvar, accessed on 5 November 2022), and the OMIM Catalogue (http://www.ncbi.nim.nih.gov/omim, accessed on 5 November 2022). In this study, we determined potentially pathogenic or “potentially causative variants” (PC-CNVs), i.e., all the CNVs defined as associated with TS in the OMIM database. Furthermore, we included in the PC-CNV category those potentially causative CNVs having a more unclear role in the disease, but sporadically associated with TS in the literature or affecting genes known to be associated with TS or other neuropsychiatric disorders. The “non-causative” CNVs (NC-CNVs) included variants of unknown significance (VOUS) that have never been reported in literature (unknown CNVs) or never described in association with TS, as well as VOUS that are likely benign or benign [30]. When a PC-CNV was detected, the study of the parents was solicited to clarify its origin and establish the strategy for family care and future genetic counselling. In some cases, the origin of the imbalances was not possible to establish, either because of indisposition of both parents to be evaluated, in situations of adoption, or in mono-parental patients in which the accessible progenitor was not a vector of the proband’s CNVs.

### 2.6. Statistical Analysis

Data were evaluated by applying SPSS software (SPSS, Inc., Chicago, IL, USA, IBM, Somers, NY, USA). Children with PC-CNV and NC-CNV and children with W-CNV were compared to examine the outcomes of the presence/absence of CNVs on dysmorphic features, comorbidities, ID, YGTSS scores, the rate of epilepsy, EEG anomalies, and brain MRI anomalies. Clinical characteristics of participants were summarized by randomized grouping using the mean (SD) for continuous data or count (%) for categorical data. Students’ t-tests were applied to compare the neuropsychological scores and characteristics between groups. A *p*-value < 0.05 was deliberated to show statistical significance.

## 3. Results

### 3.1. Genetic Analysis

Out of 93 children with TS, 20 (16 males, 4 females) tested positive for CNVs (21.5%) and 73 (55 males, 18 females) had no CNVs (78.5%). Eleven patients (9 males and 2 females) had C-CNVs (11.8%) and nine (7 males and 2 females) had NC-CNVs (9.7%). A total number of 20 imbalances were discovered (13 duplications and 7 deletions) (Table 1). From all children emerged a single CNV. Five deletions, located on chromosomes 22q11.21, 6q26, 15q11.2 (2 deletions), and 16p13.3, and six duplications, located on chromosomes 2p16.1, 22q11.21, 4p14, 7q11.23, 17q12, and 5q33.3-q34, were categorized as potentially pathogenic/causative CNVs (55% of all the CNVs) (Table 2). Nine CNVs (two deletions and seven duplications) were considered as non-causative CNVs (45.0% of all the CNVs). Among twelve rearrangements examined for the parental origin, four were paternally inherited, three were maternally inherited, and five emerged de novo.

### 3.2. Participants’ Characteristics

In this study, we recruited a total of 93 patients aged 6–18 years (mean age = 12.1 ± 3.1; male (M)/female (F) = 71:22; male = 76.3%). All patients were affected by TS. Somatic and neuropsychiatric features of subjects with TS are displayed in Table 2.

The mean age of tic onset was 6.6 ± 2.9. Of the 93 patients affected by TS, 42 subjects (45.2%) had a family history of TS, OCD, or other neuropsychiatric disease. Of all the individuals diagnosed with TS, the most frequent neuropsychiatric comorbid disorder was OCD (33.3%, n = 31). Forty-nine patients (52.7%) presented the “pure TS” phenotype. Mean full-scale IQ of all participants was 85.5 (±19.4 SD). Nineteen patients (20.4%) had IDs. Furthermore, the rate of patients with IDs was significantly different between children with C-CNVs, NC-CNVs, and W-CNVs (*p* = 0.025). Mean score for YGTSS was 20.4 (± 9.2 SD). Results on YGTSS score established no significant differences over the groups (*p* = 0.4). No significant differences were measured among groups with regard to the rate of patients with comorbidities (*p* = 0.1) (Table 2).

### 3.3. Physical Measures

Thirty-four of ninety-three children with TS (37.6%) exhibited dysmorphisms, with significant differences observed among the three groups. The number of dysmorphisms was higher in children of the group PC-CNV (63.6%), compared to NC-CNVs (44.4%) and to W-CNVs (31.5%) (*p* = 0.1) (Table 2).

### 3.4. Epilepsy and EEG Anomalies

Only seven out of ninety-three patients (7.5%) had epilepsy that required medical therapy and/or isolated EEG anomalies (focal spikes or sharp waves, slow waves). Significant discrepancies were detected among groups for the occurrence of epilepsy or isolated EEG anomalies (*p* = 0.026) (Table 2).

### 3.5. MRI Studies

MRI studies were investigated to identify morphologically visible signs of abnormal brain developmental. MRI anomalies were noted in 13 of 93 study patients (14.0%). The percentage of patients with MRI anomalies was 36.4% in the C-CNV group, and 0% and 12.3% in the NC-CNV and W-CNV groups, respectively, with high discrepancies (*p* = 0.045) (Table 2).

## 4. Discussion

The aim of this study was to identify whether the manifestation of C-CNVs is associated with typical and predictable clinical features in the TS group of patients. We systematically studied, using array-CGH, a cohort of patients with a diagnosis of primary TS, recruited prior to the study period, and considered patients with C-CNV compared to patients with NC-CNV or W-CNV. Twenty of ninety-three patients (21.5%) tested positive for CNVs. Eleven children (11.8%) had pathogenic CNV (C-CNV), of which five had a de novo mutation. Molecular and clinical and characteristics of patients with C-CNV are described in Table 3 (proband 1–20).

Among children with C-CNV, one male subject (proband 1) had a de novo duplication of the short arm of chromosome 2 (2p16.1p15 60,294,104-62,030,285) of about 1.73 Mb, encompassing many genes, including *BCL11A*, *PAPOLG*, *REL*, *PUS10*, *PEX13*, *KIAA1841*, *AHSA2*, *USP34*, *XPO1*, and *C2orf74* [31,32]. The boy, affected by TS comorbid with OCD and mild intellectual disability, also presented dysmorphic traits, including clinodactyly of the fifth finger and bilateral syndactyly of the second and third toes. This patient was already reported as a case report in 2018 [33]. A similar rearrangement was reported in a patient with a de novo interstitial microduplication, including 2p16.1-p15 (60,150,427-61,816,209) [31]. The patient reported by Mimouni-Bloch et al. (2015) presented some similar clinical characteristics, including of mild mental retardation, behavioral problems with evident OCD features, hypotonia and motor dyspraxia, and some facial dysmorphisms (puffy eyelid, broad philtrum), comparable to our Proband 1. Interestingly, this region is also associated with the 2p15p16.1 microdeletion syndrome, characterized by a complex phenotype, including neurodevelopmental delay, facial dysmorphisms, and autistic behavior [34], suggesting that this chromosomal segment may harbor dosage-sensitive genes linked to neuropsychiatric symptoms.

Proband 2, a male affected by TS associated with hypotonia, learning disabilities, and ADHD, showed a de novo microduplication of chromosome 22q11.21, a region involved in the recurrent 22q11.2 deletion and duplication syndromes, including the DiGeorge syndrome (DGS; 188,400) and velocardiofacial syndrome (VCFS; 192,430). While the 22q11.2 deletion syndrome is linked to the insurgence of neuropsychiatric problems, including intellectual disability, schizophrenia, learning disabilities, ADHD, autism spectrum disorder, anxiety, epilepsy, and early-onset Parkinson’s disease [35], the 22q11.2 microduplication syndrome shows a highly variable phenotype, encompassing asymptomatic and severe, characterized by intellectual disability, ADHD, and dysmorphic features. Furthermore, significant cardiac heart defects were reported in patients with 22q11.2 deletion and duplication syndromes [36,37]. Accordingly, our patient also presented an hypomobility of interatrial septum and mild dysmorphic features. Remarkably, Clark et al. [38] identified a 22q11.2 microduplication in a patient with TS and a Klippel-Feil anomaly.

Proband 5, a female subject affected by TS, speech disorder, and EEG abnormalities, had a partial duplication of the short arm of chromosome 4p14 of about 2.7 Mb, encompassing about 30 genes, including *TLR1* (MIM:601,194), *WDR19* (MIM:608,151), *RFC1* (MIM:102,579), *LIAS* (MIM:607,031), *UGDH* (MIM:603,370), and *RHOH* (MIM:602,037). Among them, the *RCF1* gene is associated with cerebellar ataxia, vestibular areflexia syndrome, and neuropathy (MIM:614,575) [39], while the *UGDH* gene was reported in patients affected by developmental and epileptic encephalopathy [40]. The duplication was also identified in our proband’s father, who showed intellectual disability, speech impairment, and dysmorphic features. Considering the large size of duplication, and the presence of the same duplication and psychopathologies in her affected father, we classified this variation as potentially causative.

Proband 8, a male child, presented a partial deletion of the long arm of chromosome 22 (22q11.21) of about 686 Kb, inherited by his father. The proband presented TS with comorbid OCD, a speech disorder, and minor dysmorphic traits. Furthermore, his father was affected by TS and OCD. The 22q11.21 deletion encompasses several genes, including *SCARF2* (MIM: 613,619), *PI4KA* (MIM: 600,286), *SERPIND1* (MIM:142,360), *SNAP29* (MIM:604,202), and *LZTR1* (MIM:600,574). Loss-of-function mutations in *LZTR1* increase the risk of being affected by Noonan syndrome (MIM:605,275) and an inherited disorder of multiple schwannomas [41,42]. The same CNV was already reported in a patient affected by TS and OCD in a large, genome-wide investigation of rare CNVs in OCD and TS [18].

Proband 9, a male with TS comorbid with speech delay, mild intellectual disability, OCD, ADHD, ODD, an anxiety disorder, and dysmorphic features, had a de novo partial duplication of the long arm of chromosome 7 (7q11.23) of about 1.6 Mb, which involves about 33 genes [35,36]. The chromosome 7q11.23 duplication syndrome is a multisystem developmental disease associated with variable manifestations, most commonly speech delay, intellectual disability, and mild craniofacial anomalies, including thin lips, broad forehead, abnormal columella, and deep-set eyes, such us in our proband [43,44].

Proband 12 was a boy affected by TS, intellectual disability, epilepsy, and mild dysmorphic features, carrying a 1.3 Mb paternally inherited duplication of chromosome 17q12, inherited from his father. The duplicated 17q12 region encompasses about 20 genes, including *HNF1B* (MIM:189,907), *ACACA* (MIM:200,350), *PIGW* (MIM:610,275), and *ZNHIT3* (MIM:604,500).

The 17q12 microduplication syndrome is linked to a high risk for neurodevelopmental disorders, like developmental disorders, ID (mild or severe), ASD, psychotic disorders, anxiety, and bipolar disorder [37,38]. Moreover, patients with 17q12 duplication show different phenotypes, imputable to reduced penetrance and variable expressivity [45,46].

Proband 14, a male affected by TS, OCD, a learning disorder, and dysmorphic traits like turricephaly and joint hyperlaxity, had an 805 Kb deletion of chromosome 6q26, inherited from his mother. A similar CNV was already reported in a patient affected by TS and OCD in a large, genome-wide investigation of rare CNVs in OCD and TS (McGrath et al., 2014 [18]. This deletion encompasses the *PARK2* gene (OMIM:602,544), a neurodevelopmental gene originally associated to early-onset Parkinson’s disease (PD) [47]. More recently, this gene has been associated with schizophrenia, ASD, and attention-deficit/hyperactivity disorder (ADHD) [47]. More specifically, the neurodevelopmental deletion rate was larger in OCD than in TS. Most frequent neurodevelopmental CNVs have been identified at 16p13.11 and 22q11 regions [18].

Both proband 15, a boy affected by TS, OCD, behavioral problems, and mild dysmorphic features, and proband 19, a girl affected by TS, mild intellectual disability, and ODD, presented the recurrent 15q11.2 deletion of about 395 Kb, which includes *TUBGCP5, CYFIP1, NIPA1,* and *NIPA2* genes. Some of these genes are largely expressed in the central nervous system (*CYFIP1*, *NIPA1*, *NIPA2*), while *TUBGCP5* gene is expressed in the subthalamic nuclei [48]. Neuropsychiatric disorders and mild dysmorphism with incomplete penetrance and variable expressivity are linked to genomic instabilities of the 15q11.2 BP1–BP2 region [49].

Proband 16, a male subject affected by TS, mild intellectual disability, ODD, and MRI abnormalities, had a duplication of about 2.8 Mb of chromosome 5q33.3-q34, involving several genes, including *GABRB2* (MIM:600,232), *GABRA1* (MIM:137,160), and *GABRG2* (MIM:137,164). These genes are associated with developmental and epileptic encephalopathy [50,51,52]. In this case, we could not ascertain the inheritance of the rearrangement, because of lack of both parents’ samples. To date, this imbalance is not clearly associated with a typical clinical phenotype. However, its role as a contributing cause of disease cannot be excluded.

The last proband with C-CNV, a male child with TS, severe intellectual disability, ASD, and mild dysmorphisms features, carried an interstitial deletion of chromosome of 116p13.3 inherited by his father. Loss of the 16p13.3 locus partially includes the RBFOX1, a gene encoding an RNA binding protein that is associated with a high risk of developing aggressive and antisocial behavior [53]. In addition, a recent study suggested a significant role of the RBFOX1 gene in ASD susceptibility [54].

Several studies have investigated the presence and importance of causative CNVs in TS. Almost 1% of patients with TS show one known or potentially pathogenic CNV. On the whole, chromosomal structural modifications and massive CNV increases, already known in various notable and rare diseases, have a very important role in the genetic architecture of TS [55].

In the present series, we found about 28% of patients with C-CNV with seizures and isolated epileptiform EEG anomalies, while 37% of patients with C-CNV presented qualitative anomalies on clinical MRIs. Data on the occurrence of epilepsy and EEG abnormalities in TS are very discrepant. Patients with Tourette syndrome have a higher risk of developing epilepsy than the rest of the population and are also sometimes misdiagnosed with seizures. Nowadays, not many studies have described an association between Tourette syndrome and epilepsy [56,57,58]. Rizzo et al. (2010) illustrated a case series of eight patients with Tourette syndrome with comorbid ADHD and epilepsy. The majority of these patients developed epilepsy before the onset of Tourette syndrome, even if it was controlled by medications without difficulty [58]. Wong et al. [59] showed that the incidence rate of epilepsy in the Tourette syndrome group was 4.0%, while in the control group, it was 0.2% (*p* < 0.001). On the other hand, 37% of patients with PC-CNV presented anomalies on clinical MRIs. Some studies demonstrated that neurobiological abnormalities could be a prime mover of TS and imaging studies have revealed relevant evidence [60]. It has been demonstrated that TS patients in structural imaging studies can present smaller corpus callosum volumes and thinner sensorimotor cortices, like our probands 5, 16, and 18. Furthermore, some of these regions with grey matter abnormalities may be connected with cortico-striato-thalamo-cortical (CSTC) circuits in TS.

As concerns neurobehavioral characteristics, subjects with TS holding a PC-CNV had more intense comorbid conditions, including ADHD, OCD, and ODD, than patients with NC-CNV or W-CNV. Indeed, most of the clinical variants found in our cohort are just associated with other comorbidities (e.g., ADHD, conduct disorder) rather than TS in literature studies, probably because the diagnosis of TS is often underestimated with respect to other neurological or psychiatric disorders.

The outcomes of this study also demonstrated that rare deletions and duplications highlighting significant genes for neurodevelopment had a statistically more significant rate of occurrence in children with tics and additional comorbidities. In our cohort, we determined an incidence of CNV-PC of about 12%. Our result was higher in comparison to another previous study conducted by Sundaram et al. [61] in 2010, on a cohort of 111 TS patients, which showed an incidence of about 9% of significant CNVs. Moreover, most of the CNVs identified in our TS patients, such as the 22q11.21 microduplication, the 17q12 duplication, and the 15q11.2 deletion, show incomplete penetrance and variable expressivity and are associated with different clinical presentations in the literature. Our results underline the complex genetic architecture of TS and suggest the specific prominent role of specific potentially pathogenic CNVs in the etiology of the disorder. Certainly, there are several limitations in our study. The sample size was small, giving a restricted statistical power and detection of within-group differences. Secondly, this study was conducted in a tertiary care center, in which the majority of TS patients present more comorbidities and higher tic severity. Consequently, the significant incidence of causative CNV detected in our TS sample may be correlated to selection bias of TS participants. Our results should be considered preliminary due to these limitations. Undoubtedly, further studies are essential to better delineate the genetic background of patients with tic disorders, to clarify the complex genetic structure of these disorders, to better define the outcome, and to identify new possible therapeutic targets.

## Figures and Tables

**Table 1 genes-14-00500-t001:** Demographic and genetic characteristics of TS patients.

	Patients (n)	Age (Years)	M:F Ratio	Deletions (n)	Duplications (n)
**All**	93	12.1 ± 3.1	71:22	/	/
**Patients without CNVs**	73	11.9 ± 3.2	55:18	/	/
**Patients with CNVs**	20	12.4 ± 2.8	16:4	7	13
**C-CNVs**	11	12.3 ± 2.8	9:2	5	6
**NC-CNVs**	9	12.7 ± 3.0	7:2	2	7

**Table 2 genes-14-00500-t002:** Somatic and neuropsychiatric features in patients with TS according to CNVs.

	C-CNVs (11)	NC-CNVs (9)	W-CNVs (73)	*p*-Value
**Dysmorphic features**	7 (63.6%)	4 (44.4%)	23 (31.5%)	**0.1**
**Epilepsy/EEG anomalies**	3 (27.3%)	0 (0%)	4 (5.5%)	**0.026**
**Brain MRI anomalies**	4 (36.4%)	0 (0%)	9 (12.3%)	**0.045**
**Familiarity for neuropsychiatric disorders**	7 (63.6%)	7 (77.8%)	28 (38.4%)	**0.03**
**Presence of comorbidities**	8 (72.7%)	5 (55.6%)	29 (39.7%)	0.1
**Intellectual disability**	5 (45.5%)	4 (44.4%)	10 (13.7%)	**0.01**

**IQ**	75.7 ± 26.1	74.4 ± 23.2	88.3 ± 17.0	**0.025**
**YGTSS**	17.0 ± 7.5	22.0 ± 11.7	20.7 ± 9.1	0.4

**Table 3 genes-14-00500-t003:** Clinical and molecular characteristics associated with CNVs in patients with TS. https://franklin.genoox.com/clinical-db/home. ACMG: variants of uncertain significance (VUS)—score between −0.89 and 0.89 points; likely pathogenic (LP)—score between 0.90 and 0.98 points; pathogenic (P)—score 0.99 points; likely benign (LB)—score between −0.90 and −0.98 points; benign (B)—score −0.99.

Proband	CNVs	Type of CNVs	Genes Involved	Tourette	Comorbidities	Dysmorphism
**1**	**arr[GRCh37]2p16.1-p15(60,294,104-62,030,285)x3 dn**	Potentially causative VUS (0.4)	BCL11A, CASK, USP 24 and PEX13	Onset: 7. YGTSS:12.	Mild intellectual disability. OCD. Headache. Behavior disorder.	Small forehead, thick eyebrows with horizontal orientation, low-implantation ears, hypoplasia of the tragus, short and flat nasal filter, ogival palate. Adipose tissue abundantly represented. 5-inch finger-like thumbs, clinodactyly
**2**	**arr[GRCh37]22q11.21(18,641,409-21,440,514)x3 dn**	Potentially causative P (1.00)	About 76 genes; (MIM:# 608,363)	Onset: 7. YGTSS:12.	ADHD.	Broad forehead, thick eyebrows, epicanthi, horizontal eyelid rhymes, low-implantation large ears, tragus hypoplasia, flat filter, protruding columella, ogival palate, thin upper lip, V-finger clinodactyly, finger pads
**3**	**arr[GRCh37]9q21.33(88,234,157-88,408,246)x1**	Non-causative LP (0.9)	AGTPBP1	Onset: 12. YGTSS:15.	Mild intellectual disability. OCD.	Small eyes, small forehead, hypolorism, low implantation ears, short filter, clinodactyly of the V finger, finger-like thumb, syndactyl of the II, III, and IV finger. Accentuated arch
**4**	**arr[GRCh37]16q22.1-q22.2(69,833,053-70,883,845)x3 pat**	Non-causative VUS (0.3)	WWP2, CLEC18A, PDPR, CLEC18C, EXOSC6, AARS1 (#OMIM 601,065), DDX19B, DDX19A, ST3GAL2, FUK, COG4 (#OMIM 606,976), SF3B3, IL34, MTSS1L	Onset:5. YGTSS:45.	Headache. Anxiety disorder.	Flat filter. Small nose. Tapered hands, thin fingers. Pectus excavatum. Syndactyly II-III toes
**5**	**arr[GRCh37]4p14(37,684,725-40,379,119)x3 dn**	Potentially causative VUS (0)	30 genes including TLR1, WDR19, RFC1, LIAS, UGDH and RHOH	Onset:10. YGTSS:14.	Speech disorder.	No dysmorphism
**6**	**arr[GRCh37]12q13.12(49,488,325-49,811,367)x3 mat**	Non-causative VUS (0.3)	DHH, LMBR1L, TUBA1A, TUBA1B, TUBA1C, PRPH, TROAP, C1QL4, DNAJC22, SPATS2	Onset:6. YGTSS:13.	Mild intellectual disability. OCD. Behavior disorder. Microcephaly.	Coffee-milk stains of varying sizes spread throughout the body. Broad forehead, prominent frontal drafts
**7**	**arr[GRCh37]Xp22.31(6,552,712-8,115,153)x2**	Non-causative B (−1)	HDHD1, STS, VCX, PNPLA4 (X)	Onset:7. YGTSS:15.	No	Large eyes with low implantation, sparse eyebrows, horizontal eyelid rhymes, hypolorism. Short filter, ogival palate, clinodactyly of the V finger, mild hypotonia, II and III toe syndactyly bilaterally, finger-like thumbs.
**8**	**arr[GRCh37]22q11.21(20,754,422-21,440,514)x1 pat**	Potentially causative P (1.00)	ZNF74, SCARF2 (MIM: 613,619), KLHL22, MED15, PI4KA (MIM: 600,286), SERPIND1 (MIM:142,360), SNAP29 (MIM:604,202), CRKL, AIFM3, LZTR1 (MIM:600,574), THAP7, P2RX6, SLC7A4	Onset:2. YGTSS:34.	OCD. Behavior disorder. Speech disorder.	Large eyes with low implantation, sparse eyebrows, horizontal eyelid rhymes, hypolorism. Short filter, ogival palate, clinodactyly of the V finger, mild hypotonia, II and III toe syndactyly bilaterally, finger-like thumbs.
**9**	**arr[GRCh37]7q11.23(72,645,480-74,197,434)x3 dn**	Potentially causative P (1.00)	33 genes	Onset:6. YGTSS:20.	Mild intellectual disability. OCD. ADHD. ODD. Anxiety disorder.	Turricephaly, high forehead, epicanthi, hypolorism, horizontal eyelid rhymes, pointed columella, microretrognathia, thin lips, chin dimple, ogival palate, clinodactyly of the V finger, winged scapulae.
**10**	**arr[GRCh37]10q22.3(78,651,963-79,327,368)x3 pat**	Non-causative VUS (0.45)	KCNMA1 (#MIM 600,150)	Onset:11. YGTSS:9.	Mild intellectual disability. OCD.	No dysmorphism
**11**	**arr[GRCh37]6q22.31(123,539,625_124,280,442)x3**	Non-causative VUS (0.15)	TRDN (#OMIM 603,283)	Onset:4. YGTSS:20.	Anxiety disorder.	No dysmorphism
**12**	**arr[GRCh37]17q12(34,817,422-36,168,104)x3 pat**	Potentially causative P (1.00)	20 genes: HNF1B (MIM: 189,907), ACACA (MIM:200,350), PIGW (MIM:610275), ZNHIT3 (MIM:604,500)	Onset:13. YGTSS:10.	Epilepsy.	Small forehead. Symphysis. Hypolorism. Poorly shaped and low-implanted ears. Horizontal eyelid rhymes. Prominent columella. Short filter. Epitrochlear dimples. Pointed palate. Finger-like thumbs
**13**	**arr[GRCh37]Xq28(148,798,762-149,572,551)x2**	Non-causative VUS (0.3)	MAGEA11, HSFX1, MAGEA9B, MAGEA9, MAGEA8, CXorf40B, MAMLD1 (#OMIM 300,120)	Onset:6. YGTSS:25.	Severe intellectual disabilities. Behavior disorder.	No dysmorphism
**14**	**arr[GRCh37]6q26(161,878,327-162,683,777)x1 mat**	Potentially causative P (1.00)	PARK2 (#OMIM 602,544)	Onset:3 YGTSS:19.	OCD. Specific learning disorder.	No dysmorphism
**15**	**arr[GRCh37]15q11.2(22,784,523-23,179.948)x1**	Potentially causative P (1.00)	TUBGCP5, CYFIP1, NIPA2, NIPA1 (MIM:608,145)	Onset:2. YGTSS:11.	OCD.	Mild turricephaly, sparse eyebrows, poorly structured ears, mild bilateral gynecomastia and inverted nipples. Pterygia of the neck.
**16**	**arr[GRCh37]5q33.3-q34(158,829,101-161,658,146)x3**	Potentially causative VUS (0)	ADRA1B, TTC1, PWWP2A, FABP6, CCNJL, C1QTNF2, C5orf54, SLU7, ATP10B, PTTG1, GABRB2 (#OMIM 600,232), GABRA6, GABRA1 (#OMIM 137,160), GABRG2 (#OMIM 137,164)	Onset:6. YGTSS:10.	Mild intellectual disability. ODD. Behavior disorder.	No dysmorphism
**17**	**arr[GRCh37]16p12.2(21,599,687-21,739,885)x1 mat**	Non-causative LP (0.90)	OTOA (MIM:607,038)	Onset:7. YGTSS:36.	Mild intellectual disability. Behavior disorder. Microcephaly.	No dysmorphism
**18**	**arr[GRCh37]16p13.3(6,880,891-7,008,524)x1 pat**	Potentially causative VUS (-0.21)	RBFOX1	Onset:7. YGTSS:25.	Severe intellectual disabilities. Disorder of the autistic spectrum.	Flat front on the right, low-implantation ears, horizontal eyelid rhymes. Right eyelid ptosis. Excavated chest. V-finger clinodactyly.
**19**	**arr[GRCh37]15q11.2(22,784,523-23,179.948)x1**	Potentially causative P (1.00)	TUBGCP5, CYFIP1, NIPA2, NIPA1 (MIM:608,145)	Onset:5. YGTSS:20.	Mild intellectual disability. ODD.	No dysmorphism.
**20**	**arr[GRCh37]4q26(119,461,777-119,830,456)x3**	Non-causative VUS (0)	METTL14, SEC24D (#OMIM 607,186), SYNPO2	Onset:3. YGTSS:20.	No	No dysmorphism

## Data Availability

The data presented in this study are available on request from the corresponding author.

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
