# Peer review of "Copy Number Variations in Children with Tourette Syndrome: Systematic Investigation in a Clinical Setting"

_genes, 2023, doi:10.3390/genes14020500_

Round 1

Reviewer 1 Report

This is an interesting study that reports some positive findings in CNV analyses of 93 well characterized patients with Tourette syndrome. A few comments: 

1. First of all, any "positive" finding should be defined as potentially "causative" or "pathogenic" other than "causative-CNV" directly. Without further experimental evidence and/or additional clinical cases, it's hard to establish the causal relationship between a positive finding from CNV-CGH assay and a clinical diagnosis of TS.

2. Definitions of "causative-CNV", and "non-causative CNV" should be aligned with other standardized studies in the literature in order to make the results comparable to other studies, e. g.,ACMG and ClinGen standards and criteria on pathogenic CNVs.

3. There was no detailed description of clinical assessments of neuropsychiatric comorbidities, such as ASD, OCD and ADHD etc.

4. There was no detailed description on how the statistical analyses were performed exactly among different groups. The CNV negative groups were more likely due to the technology limitation, i.e., the resolution of CGH array, other than truly negative for CNVs.  How about multiple testing corrections?   

Author Response

Reviewer 1

Point1: First of all, any "positive" finding should be defined as potentially "causative" or "pathogenic" other than "causative-CNV" directly. Without further experimental evidence and/or additional clinical cases, it's hard to establish the causal relationship between a positive finding from CNV-CGH assay and a clinical diagnosis of TS.

Response 1: We thank the reviewer for all useful comments and suggestions. We are extremely grateful for your time spent reviewing our article. The following are our answers point by point to your issues.

First of all, I fixed "C-CNV" with Potentially Causative CNV in the main text.

Point 2.Definitions of "causative-CNV", and "non-causative CNV" should be aligned with other standardized studies in the literature in order to make the results comparable to other studies, e. g., ACMG and ClinGen standards and criteria on pathogenic CNVs.

Response 2: I also added the nomenclature of the CNV following the ACMG guidelines, using Franklin's database

Point 3.There was no detailed description of clinical assessments of neuropsychiatric comorbidities, such as ASD, OCD and ADHD etc.

Response 3: Regarding the description of clinical assessment of other comorbidities, I added more details. All patients were screened with the Schedule for affective disorders and Schizophrenia for School age children—present and lifetime (Kiddie-SADS-PL) (Kaufman et al., 1997) to evaluate other possible comorbidities. Then, all patients underwent neuropsychiatric evaluation for TS and related comorbidities. Symptoms of TS were evaluated using the Yale Global Tic Severity Rating Scale (YGTSS) (Leckman et al., 1989), a clinician-rated scale used to evaluate the motor and phonic tic severity in the light of the number, frequency, duration, intensity, and complexity of tics. To evaluate obsessive-compulsive disorder (OCD), commonly associated with TS or CTD, the CY-BOCS, a semi-structured clinician-administered interview assessing the severity of obsessions and compulsions occurring over the past week across five areas (time, interference, distressing nature, effort to resist, control over obsessions and compulsions) was also administered (Scahill et Al., 1997). Regarding ADHD symptoms, patients were also assessed according to the ADHD rating scale (Pappas et Al., 2006) and the Conners’ Parent Rating Scale (CPRS). The CPRS is a useful tool for obtaining parental reports of childhood behavior problems that contains summary scales supporting ADHD diagnosis and quantifying ADHD severity (Conners C.K, 1997). Diagnosis of ASD was evaluated using gold-standard standardized diagnostic tests including the Autism Diagnostic Interview-Revised (ADI-R) and Autism Diagnostic Observation Schedule (ADOS). ADI-R is an investigator-based parent or caregiver interview that yielded a description of history, as well as current functioning, in areas of development related to autism (Lord et Al., 1994). ADOS is a semi structured, standardized assessment of social affect, which comprehends language, communication and social reciprocal interaction, and restricted and repetitive behaviors for individuals suspected of having ASD (Gotham et Al., 2009). Finally, all participants completed the MASC, a self-report scale that robustly represents the factor structure of anxiety in children aged 8-18 years (March et al., 1997) and the Child Depression Inventory: a 27-item self-report instrument that assesses depressive symptoms in 7- to 17-year-olds (Kovacs M., 1988).

Point 4.  There was no detailed description on how the statistical analyses were performed exactly among different groups. The CNV negative groups were more likely due to the technology limitation, i.e., the resolution of CGH array, other than truly negative for CNVs.  How about multiple testing corrections?   

Response 4: Regarding the different groups, with negative and positive CNVs, the resolution of the CGH Array was the same, i.e. we used Agilent Reference DNAs, analyzed with the Agilent Microarray Scanner Feature Extraction Software version 11.5, and Agilent Genomic Workbench 7.0. 4.0 software using the ADM-2 algorithm. Genomic positions of the rearrangements refer to the public UCSC database GRCh37/hg19. Furthermore, we have to clarify that our  study was conducted in a tertiary-care centre, where the majority of TS patients have higher comorbidity and tic severity. Therefore, the higher incidence of causative CNV detected in our TS sample may be correlated to selection bias of TS participants. Moreover, we added in the discussion section the limitations of our study, and that our results should be considered preliminary, due to these limitations.

Reviewer 2 Report

The study is aimed to define the neurobehavioral phenotype of patients with or without pathogenic Copy Number Variation (CNV), and to compare significant CNVs with CNV described in the literature in neuropsychiatric disorders, including TS, to describe an effective clinical and molecular characterization of patients for prognostic purposes and for the correct taking charge. Besides, this study showed that rare deletions and duplications highlighting relevant genes for neurodevelopment had a statistically higher occurrence in children with tics and additional comorbidities. The authors found the incidence of CNV-C of about 12%, in line with other literature studies.

My comment is that there are several limitations of the studies. First, the numbers of this study are small as to make them questionably significant. Second, the incidence of CNV-C are higher than most literature studies, and it may be due to selection bias of their patients, since most TS patients have no intellectual disability. The results of CNV are variable, and the interpretation of the CNV data is not clear, such as their relevance of TS pathologic mechanism.

In conclusion, this is an important issue to understand genetic factors of TS, since CNV data of TS is few in recent studies, and CNV is a popular tool of genetic tool in most countries. The authors reminds us applying this genetic test if the occurrence of dysmorphic features, epilepsy, brain Magnetic Resonance Imaging anomalies, and severity of symptoms in children with TS 

Author Response

Reviewer 2

My comment is that there are several limitations of the studies. First, the numbers of this study are small as to make them questionably significant. Second, the incidence of CNV-C are higher than most literature studies, and it may be due to selection bias of their patients, since most TS patients have no intellectual disability. The results of CNV are variable, and the interpretation of the CNV data is not clear, such as their relevance of TS pathologic mechanism.

In conclusion, this is an important issue to understand genetic factors of TS, since CNV data of TS is few in recent studies, and CNV is a popular tool of genetic tool in most countries. The authors remind us applying this genetic test if the occurrence of dysmorphic features, epilepsy, brain Magnetic Resonance Imaging anomalies, and severity of symptoms in children with TS 

We thank the reviewer for all useful comments and suggestions. We are extremely grateful for your time spent reviewing our article. Moreover, we followed your suggestion and clarified in the main text the limitations of the study. Certainly, there are several limitations in our study. First, the sample size was small, limiting statistical power and detection of within-group differences. Second, our study was conducted in a tertiary-care centre, where the majority of TS patients have higher comorbidity and tic severity. Therefore, the higher incidence of causative CNV detected in our TS sample may be correlated to selection bias of TS participants. Following your advice, we have specified in the discussion section as our results should be considered preliminary, due to these limitations.

Round 2

Reviewer 1 Report

The manuscript has been significantly improved based the comments from the first reviewer.